# Strategies for Effective Management of Indoor Air Quality in a Kindergarten: CO_2_ and Fine Particulate Matter Concentrations

**DOI:** 10.3390/toxics11110931

**Published:** 2023-11-16

**Authors:** Doyeon Lee, Younghun Kim, Kee-Jung Hong, Gunhee Lee, Hak-Joon Kim, Dongho Shin, Bangwoo Han

**Affiliations:** 1Department of Sustainable Environment Research, Korea Institute of Machinery & Materials, Daejeon 34103, Republic of Koreadiayolk@kimm.re.kr (H.-J.K.); 2Department of Mechanical Engineering, University of Science and Technology, Daejeon 34113, Republic of Korea; 3Department of Mechanical Engineering, Yonsei University, Seoul 03722, Republic of Korea

**Keywords:** kindergarten, CO_2_, PM_2.5_, mechanical ventilation, indoor air quality, numerical model

## Abstract

The educational and play-related activities of children proceed mainly indoors in a kindergarten. High concentrations of indoor PM_2.5_ and CO_2_ have been linked to various harmful effects on children, considerably impacting their educational outcomes in kindergarten. In this study, we explore different scenarios involving the operation of mechanical ventilation systems and air purifiers in kindergartens. Using numerical models to analyze indoor CO_2_ and PM_2.5_ concentration, we aim to optimize strategies that effectively reduce these harmful pollutants. We found that the amount of ventilation required to maintain good air quality, per child, was approximately 20.4 m^3^/h. However, we also found that as the amount of ventilation increased, so did the concentration of indoor PM_2.5_; we found that this issue can be resolved using a high-grade filter (i.e., a MERV 13 grade filter with a collection efficiency of 75%). This study provides a scientific basis for reducing PM_2.5_ concentrations in kindergartens, while keeping CO_2_ levels low.

## 1. Introduction

Indoor air quality management is essential, especially since people spend more than 80% of their lives indoors [1,2]. According to a report from the World Health Organization (WHO), indoor PM_2.5_ (fine particulate matter) was responsible for approximately 2.3 million deaths in 2020 [3]. In particular, children exposed to high PM_2.5_ levels have been shown to manifest a variety of negative health-related effects (e.g., pneumonia, high blood pressure, and heart disease) [4,5,6]. The respiration rate in children is relatively higher than that in adults when considering intake per unit body weight. Consequently, children are more vulnerable to the adverse effects of airborne toxic substances [6]. 

Indoor air pollutants can pose significant health risks to humans. Formaldehyde, emitted from adhesives in furniture and wallpapers, is a well-known indoor toxic gas that can cause symptoms associated with sick building syndrome [7]. Combustion gases released from gas stoves contain harmful substances such as carbon monoxide and nitrogen dioxide. Carbon monoxide can interfere with oxygen transport, leading to poisoning and even death [8]. Nitrogen dioxide can cause respiratory diseases [9]. Hydrogen sulfide is highly toxic and can cause irritation to the skin [10]. One of the prominent indoor toxic gases, radon, is prone to emanating from sources such as latex mattresses and furniture. Prolonged exposure to radon can lead to lung cancer and respiratory diseases [11]. Various studies have substantiated the harmful effects of these indoor toxic gases, and ventilation is primarily advocated as a solution [12,13]. However, managing such a diverse array of hazardous gases individually poses a challenge. Hence, CO_2_, which is relatively easy to measure, has been employed as an indicator of indoor toxic gases [14]. 

However, increasing ventilation rates indiscriminately to mitigate indoor toxic gases is not advisable. This is because higher ventilation rates can lead to an influx of external PM_2.5_ particles, causing an increase in indoor PM_2.5_ concentrations. Moreover, excessive ventilation can result in the reduced effectiveness of air purifiers, making it challenging to manage indoor PM_2.5_ concentrations during ventilation [15]. Therefore, it is necessary to develop strategies that simultaneously consider CO_2_ and PM_2.5_ concentrations when managing indoor air quality. The Ministry of Environment in Korea has stipulated 1000 ppm of CO_2_ as the standard to maintain good air quality in kindergartens [16], whereas the WHO recommends that average indoor PM_2.5_ concentrations should be less than 15 μg/m^3^; per day [3].

In four kindergartens surveyed in Korea, their indoor CO_2_ concentrations exceeded 1000 ppm, respectively [17]. This sample of kindergartens highlights the necessity of determining the indoor ventilation rate that can maintain indoor CO_2_ below 1000 ppm. In addition, previous studies have shown that indoor PM_2.5_ concentrations, on average, exceed 15 μg/m^3^; daily in various indoor spaces (e.g., classrooms and offices) [18,19]. 

Han et al. (2022) investigated indoor PM_2.5_ variations resulting from air purifier usage in an elementary school [19]. Noh and Yook (2016) used a numerical model to determine how to reduce indoor 0.3 and 3 μm particle concentrations using mechanical ventilation filters and air purifiers [20]. Nevertheless, these studies failed to provide a viable solution for reducing indoor CO_2_ concentrations. 

Peter et al. (2000) emphasized the importance of adequate ventilation in reducing indoor CO_2_ concentration in middle schools [21]. Similarly, Franco et al. (2020) explored optimal ventilation methods while focusing on CO_2_ concentration in university classrooms [22]. However, both studies did not account for indoor particulate matter. It is essential to note that increasing ventilation rates may inadvertently lead to higher concentrations of outdoor particulate matter infiltration.

Pacitto et al. (2020) studied the changes in indoor CO_2_ and particulate matter in four different scenarios with or without window opening and air purifier operation [23]. In this study, only the measured data and the required ventilation rates were presented; there are no results regarding the expected concentration changes based on the provided ventilation rates. 

Children are particularly vulnerable to particulate matter; thus, effective indoor air quality management in kindergartens is crucial. Nevertheless, there is a limited body of research addressing indoor air quality improvement specifically in kindergarten settings. Park et al. (2017) only measured the air quality (CO_2_, PM_10_) in kindergartens in Korea [17]. Nicolas et al. (2013) conducted a study measuring formaldehyde and benzene levels in a French kindergarten [24]. However, these studies did not suggest management methods for indoor air quality in kindergartens.

This study presents methods for simultaneously managing indoor CO_2_ and PM_2.5_ concentrations in a kindergarten using numerical analysis modeling established through measurements. CO_2_ and PM_2.5_ concentrations were measured throughout the day in the kindergarten and equations were developed to model the concentration changes over time within the test space. The accuracy of the equations was verified by comparing the measured concentrations with the concentrations obtained by numerical model. Overall, through the use of numerical modeling and experimental measurements, the appropriate mechanical ventilation rate to maintain CO_2_ levels below 1000 ppm was suggested. In addition, to maintain low PM_2.5_ level under such a ventilation rate, the application of a high-grade filter in a mechanical ventilation system and a high-capacity air purifier were analyzed.

## 2. Materials and Methods

### 2.1. Experimental Setup 

This study was conducted from January to March 2023, which is the winter season in Korea. The outdoor PM_2.5_ concentrations in winter were higher than those in other seasons [25], potentially resulting in increased indoor PM_2.5_ concentrations due to the inflow of outdoor PM_2.5_ [26].

Figure 1 shows the floor plan of the experimental setup in a kindergarten in Daejeon, South Korea. The test was conducted in two classrooms. Both classrooms have the same area of 57.42 m^2^. In Classroom 1, which contained four windows, two air purifiers were arranged and the mechanical ventilation diffusers were equipped with two air supplies and two exhaust outlets. In Classroom 2, which featured three windows, a single air purifier was placed, as was a mechanical ventilation system similar to that in Classroom 1.

To prevent children from accidentally handling a device or breaking it, the measurement devices were installed on shelves at a height of 1.5 m. In practice, the doors remain closed during class; therefore, measurement errors caused by the doors are nearly negligible. The CO_2_ concentrations were measured using an NDIR-type sensor (GTH53, EYC-tech, New Taipei City, Taiwan). The measuring range and linear accuracy of the CO_2_ sensor is from 0 to 5000 ppm and ±50 ppm, respectively. For measuring PM_2.5_ concentrations, an optical particle counter (OPC, 1.109, Grimm Aerosol Technik Co., Ainring, Germany) was used. The OPC uses the light scattering method to measure the particle concentration. The measurable size range of OPC is from 0.25 to 32 μm, and the sampling flow rate is 1.2 L/min. All the measurement devices used in this test were calibrated. Measurements were performed throughout the day in the kindergarten. Measurement devices were installed in Classrooms 1, 2, and outside, recording CO_2_ and PM_2.5_ concentration data at 1 min intervals. Additionally, the power consumption of ventilators and air purifiers were monitored to ensure their proper operation. The obtained parameters were utilized in the numerical analysis equation, and the measured data were then compared with the results from the numerical analysis.

### 2.2. Numerical Model 

Figure 2 shows a schematic of the factors influencing indoor CO_2_ concentrations and indoor PM_2.5_. Based on this schematic, a mass balance equation was formulated to describe the variations in indoor CO_2_ and PM_2.5_ concentrations.

Figure 2a presents a diagram of the factors that affect the indoor CO_2_ concentration. These factors can be categorized into natural ventilation, mechanical ventilation, and human respiration. Here, natural ventilation is a method of supplying air to the classroom through the wall by opening the windows. Mechanical ventilation is a method of supplying the air through the mechanical ventilation system. In this study, balanced ventilation was used. Here, $C_{{CO}_{2},out}$ represents the outdoor CO_2_ concentration (ppm), and $C_{{CO}_{2}}$ represents the indoor CO_2_ concentration (ppm). $Q_{inf}$ represents the air inflow through the walls and windows (m^3^/h), and $Q_{exf}$ represents the air outflow through the walls and windows (m^3^/h). $Q_{MV,SA}$ represents the air inflow through the mechanical ventilation system (m^3^/h), and $Q_{MV,EA}$ represents the air outflow through the mechanical ventilation system (m^3^/h). V represents the volume of the indoor area (m^3^).

The parameter G represents the amount of CO_2_ generated through respiration (ppm × m^3^/h) and was determined using Equation (1). In this equation, “Man”, “Woman”, and “Child” denote the number of adult men, adult women, and children indoors, respectively. Based on a study conducted by Cho et al. [27], the CO_2_ generation rate through an adult man’s respiration is 305.3 ppm × m^3^/min, while it is 264.2 ppm × m^3^/min for adult women. Additionally, the study found that children generate approximately 73% of the CO_2_ produced by adults [27]. This study assumes that each child produces the same amount of generated CO_2_. Hence, in this study, the CO_2_ generation rate in the children was determined as 268.9 × 0.73 ppm × m^3^/min. The indoor CO_2_ concentration over time is given by Equation (2).
(1)$$G=\left[Man\times 305.3+Woman\times 264.2+Child\times 268.9\times 0.73\right]$$
(2)$$C_{{CO}_{2}}\left(t\right)=\frac{C_{{CO}_{2},out}\left(Q_{inf}+Q_{MV,SA}\right)+G}{Q_{exf}+Q_{MV,EA}}+\left[C_{{CO}_{2}}\left(0\right)-\frac{C_{{CO}_{2},out}\left(Q_{inf}+Q_{MV,SA}\right)+G}{Q_{exf}+Q_{MV,EA}}\right]\times {exp}^{-\frac{Q_{MV,SA},+Q_{inf}}{V}\;t}$$

Figure 2b shows the factors affecting indoor PM_2.5_ concentration. These factors can be categorized into natural ventilation, mechanical ventilation, air purifiers, and deposition. $C_{{PM}_{2.5},out}$ represents the outdoor PM_2.5_ concentration (μg/m^3^), while $C_{{PM}_{2.5}}$ signifies the indoor PM_2.5_ concentration (μg/m^3^). $\eta_{inf}$ represents the collection efficiency through natural ventilation (−), $\eta_{MV}$ represents the collection efficiency of the mechanical ventilation system filter (−). $\eta_{AP}$ represents the collection efficiency of the filter installed in an air purifier (−) and $Q_{AP}$ represents the flowrate of an air purifier (m^3^/h). So, $\eta_{AP}Q_{AP}$ represents the stated clean air delivery rate (CADR) of an air purifier (m^3^/h). ε is the air mixing factor of an air purifier in a space [28]. $\dot{S}$ represents the deposition rate (h^−1^). The indoor PM_2.5_ concentration according to time is shown in Equation (3).
(3)$$C_{{PM}_{2.5}\;}\left(t\right)=C_{{PM}_{2.5},out}\frac{\left(1-\eta_{inf}\right)Q_{inf}+\left(1-\eta_{MV}\right)Q_{MV,SA}}{Q_{exf}+Q_{MV,EA}+\dot{S}V+\varepsilon \eta_{Ap}Q_{Ap}}+\left[C_{{PM}_{2.5}\;}\left(0\right)-C_{{PM}_{2.5},out}\frac{\left(1-\eta_{inf}\right)Q_{inf}+\left(1-\eta_{MV}\right)Q_{MV,SA}}{Q_{exf}+Q_{MV,EA}+\dot{S}V+\varepsilon \eta_{Ap}Q_{Ap}}\right]\times \;{exp}^{-\frac{Q_{inf}+Q_{MV,SA}+\dot{S}V+\varepsilon \eta_{Ap}Q_{Ap}}{V}\;t}$$

### 2.3. Defining the Parameters

The parameters essential for calculating the numerical model from Equations (2) and (3) were applied as follows: The values of $C_{{CO}_{2},out}$ and $C_{{PM}_{2.5},out}$ were derived from the measured data obtained from the measurement device placed outdoors. The value of V, which was the same in Classrooms 1 and 2, was determined to be 143.55 m^3^. G was determined by the number of occupants in the indoor space and substituting these values into Equation (1). $\dot{S}$ was set to 0.05 h^−1^ based on prior research [15,29].

As shown in Figure 3, the indoor CO_2_ concentration was measured over time to determine the value of $Q_{inf}$. The measurement for $Q_{inf}$ was conducted on 7 November 2022, with an outdoor wind speed of 1.7 m/s and a temperature difference of 21 °C between the indoor and outdoor environments. As shown in Figure 3a, the initial indoor CO_2_ concentration was set to 6000 ppm with all windows and doorways closed. Based on the results shown in Figure 3a, $Q_{inf}$ was calculated to be 26 m^3^/h for Classrooms 1 and 16 m^3^/h for Classrooms 2. The higher $Q_{inf}$ value for Classroom 1 compared to Classroom 2 can be attributed to the greater number of windows in Classroom 1, leading to potentially lower airtightness [30]. As shown in Figure 3b, the indoor CO_2_ concentration was measured for all open windows. According to the data in Figure 3b, the calculated $Q_{inf}$ values were 571 m^3^/h for Classrooms 1 and 207 m^3^/h for Classrooms 2. The study conducted by Cho et al. (2012) indicated that the natural ventilation rate tended to be proportional to the opening area [30]. As the area of the openings per window remained constant, $Q_{inf}$ for Classroom 1 was determined to be 143 m^3^/h per open window, and for Classroom 2, it was determined to be 69 m^3^/h per opened window.

$Q_{MV,SA}$ were determined through airflow measurements in a diffuser using a flowmeter (Model 6750, KANOMAX, Osaka, Japan). The measurable flowrate is from 8 to 600 m^3^/h and the accuracy is ±3% from 8 to 350 m^3^/h, and ±5% from 350 to 600 m^3^/h. The mechanical ventilation system can be operated at two different flow rates, Levels 1 and 2. The airflow rates at each level were 142 m^3^/h and 230 m^3^/h in Classroom 1 and 157 m^3^/h and 254 m^3^/h in Classroom 2. The value of $\eta_{MV}$ is 0.35, which has a filter grade of MERV 11. To satisfy the continuity equation, it was assumed that the total inflow into the classroom and total outflow were equivalent. This assumption is expressed through Equation (4).
(4)$$Q_{inf}+Q_{MV,SA}=Q_{exf}+Q_{MV,EA}$$

Figure 4 shows the variation of indoor PM_2.5_ concentrations over time, with all windows and doorways closed. This approach allowed us to determine the in/out ratios. As time, *t* approaches infinity in Equation (3), the term ${exp}^{-\frac{Q_{inf}+Q_{MV,SA}+\dot{S}V+\varepsilon \eta_{Ap}Q_{Ap}}{V}\;t}$ becomes negligible, yielding the expression provided in Equation (5). This outcome enables us to derive the value of $\eta_{inf}$ as mentioned in Equations (3) and (5). Through this analysis, the values of $\eta_{inf}$ for Classroom 1 and Classroom 2 were determined as 0.404 and 0.562, respectively.
(5)$$\frac{C_{{PM}_{2.5}\;}}{C_{{PM}_{2.5},out}}=\frac{\left(1-\eta_{inf}\right)Q_{inf}+\left(1-\eta_{MV}\right)Q_{MV,SA}}{Q_{exf}+Q_{MV,EA}+\dot{S}V+\varepsilon \eta_{Ap}Q_{Ap}}$$

To obtain the value of ${\eta_{Ap}Q}_{Ap},$ we conducted measurements in a 30 m^3^ chamber using the standard test of the stated CADR (SPS-KACA002-0132 [31]). The air purifier within the kindergarten can be operated across flow rates of levels 1–4. The stated CADR of each level were 120, 210, 324, and 480 m^3^/h. However, the actual CADR measured from the classroom differ from the stated CADR measured from standard chamber. To determine the actual CADR, ε must be applied to account for the air circulation rate when the air purifier is in operation [20,28,29]. Figure 5 shows the ε values observed in both Classroom 1 and Classroom 2. These measurements were conducted for 30 min and normalized to the initial concentration. During the measurements, one air purifier was operated in Classroom 1, whereas two air purifiers were operated in Classroom 2. All air purifiers were operated at level 2. The ε was calculated as 0.75 for Classroom 1 and 0.5 for Classroom 2. Given that the two air purifiers were situated at different locations within Classroom 1, it can be concluded that the air circulation rate in Classroom 1 was superior to that of Classroom 2.

## 3. Results

### 3.1. Comparing the Numerical Model with Measured Data

In this study, we used two specific days within the January to March 2023 timeframe to compare the concentrations of indoor CO_2_ and PM_2.5_ and those obtained via the numerical models. Malik (1978) demonstrated that natural ventilation is influenced by variables such as outdoor wind speed and the temperature differential between outdoor and indoor environments [32]. As such, 3 January 2023 and 11 January 2023, were selected because of their similarities to the outdoor wind speed and temperature difference observed on 7 November 2022, which served as a reference point in Figure 3.

Table 1 shows various parameters, including the number of occupants, $Q_{inf}$, $Q_{MV}$, and ${\eta_{Ap}Q}_{Ap}$ for both Classroom 1 and 2. The information presented in Table 1 was recorded during two distinct time periods: from 9:00 to 15:00, which encompassed the class time of the kindergarten, and 15:00–15:30, when the children left the premises. On 3 January 2023, the mechanical ventilation system operated at level 2, whereas on 11 January 2023, the system operated at level 1.

#### 3.1.1. The Concentrations of CO_2_

Figure 6 shows comparisons between the measured data of the indoor CO_2_ concentrations and those obtained from the numerical model according to the time of day. Specifically, the graphs labeled as ‘In the Classroom 1’ and ‘In the Classroom 2’ represent the measured CO_2_ concentration. On the other hand, the data labeled as ‘Numerical Model’ were obtained by substituting the recorded values from Table 1 into Equation (2) for CO_2_ concentration.

Figure 6a presents the comparison data for Classroom 1 on 3 January 2023. Between 9:00 and 9:30, indoor CO_2_ concentrations gradually increased to approximately 815 ppm as the children arrived at the kindergarten. During the history class from 9:30 to 10:30, characterized by a passive learning environment, the windows were closed, leading to a rise in CO_2_ concentration to approximately 1105 ppm. Between 10:30 and 12:20, during which the children engaged in outdoor activities, indoor CO_2_ concentrations considerably decreased to 585 ppm. From 12:20 to 14:50, there was lunchtime and playtime in the classroom, during which the children’s activity levels increased sharply, and the CO_2_ concentration increased to 1180 ppm. Consequently, there were discrepancies between the measured values and the numerical analysis results in Figure 6a. This can be attributed to the rapid increase in activity levels, which were not accurately reflected in the numerical analysis results. At 14:50, when the children went home, there was a decrease in CO_2_ concentrations.

Figure 6b presents the comparison data for Classroom 2 on 3 January 2023. From 9:00 to 9:50, indoor CO_2_ concentrations gradually increased to approximately 740 ppm as the children arrived at the kindergarten. During the history class from 9:50 to 11:00. The CO_2_ concentration in Classroom 2 only increased to 1090 ppm. There were discrepancies between the measured values and numerical analysis results for 10:50–11:00; we assumed that there was high-level indoor activity as children were gearing up to go outside. Between 11:00 and 12:40, the children of Classroom 2 also engaged in outdoor activities, resulting in a considerable decrease in indoor CO_2_ concentrations to 600 ppm. Lunchtime and naptime was from 12:40 to 15:10, unlike the Classroom 1. Therefore, there was no significant discrepancy between measured and numerical values, and the indoor CO_2_ concentration only increased to 1005 ppm. From 15:10, the children went home, and the windows were opened, decreasing CO_2_ concentrations. 

Figure 6c shows the comparison data for Classroom 1 on 11 January 2023. From 9:00 to 9:40, indoor CO_2_ concentrations gradually increased to approximately 870 ppm as the children arrived at the kindergarten. From 9:40 to 12:00, the children engaged in outdoor activities. From 9:40 to 11:00, one window was opened. Consequently, the indoor CO_2_ concentrations decreased sharply, reaching approximately 565 ppm and converging around this concentration. From 11:00 to 12:00, the CO_2_ concentrations slightly increased because of the closed window and the two teachers who remained in the classroom. From 12:00 to 15:00, there was a lunchtime and roleplay activity session with one window open in Classroom 1. The increased number of children initially led to a rise in CO_2_ concentrations. However, there was an increase in CO_2_ concentration initially to 1050 ppm after which the windows were opened and this concentration did not increase further. At 15:00, the children went home, and all the windows were opened, leading to decreased CO_2_ concentrations.

Figure 6d shows the comparison data for Classroom 2 on 11 January 2023. From 9:00 to 9:30, indoor CO_2_ concentrations gradually increased to approximately 640 ppm as the children arrived at the kindergarten. Most of the children had completed their arrival by 9:30, and they watched educational videos from 9:30 to 11:00 in Classroom 2 with all the windows closed. During this time, the CO_2_ concentrations increased up to 1050 ppm. From 11:00 to 12:30, the children engaged in outdoor activities, and despite the presence of two teachers in the classroom, the indoor CO_2_ concentration decreased to 790 ppm through mechanical ventilation. From 12:30 to 15:00, there was lunchtime and naptime. The indoor CO_2_ concentration increased to 1230 ppm. Compared to Figure 6b, during the same period with a similar number of people and all the windows closed, the indoor CO_2_ concentration was higher. This was due to a decrease in $Q_{MV}$ from 254 to 157 m^3^/h. From 15:00, the children went home, and the windows were opened, leading to decreased CO_2_ concentrations.

Figure 6 shows that, except for when the children’s CO_2_ generation increased due to high activity levels, the measured values and numerical analysis results were nearly the same, with an error rate of 6%. By analyzing Figure 6b,d, it was observed that increasing the ventilation system’s airflow rates could also maintain lower indoor CO_2_ concentrations. 

#### 3.1.2. The Concentrations of PM_2.5_


Figure 7 shows a comparison between the measured data of indoor PM_2.5_ concentrations and those obtained from the numerical model according to the time of day. The data labeled ‘Numerical Model’ were obtained by substituting the recorded conditions from Table 1 into Equation (3) for PM_2.5_ concentrations. 

Figure 7a shows the comparison data on PM_2.5_ concentrations for Classroom 1 on 3 January 2023. The windows were closed throughout the day, and the flow rates of mechanical ventilation and the air purifier were constant. Unlike Figure 6a, indoor PM_2.5_ concentrations vary with outdoor PM_2.5_ concentrations regardless of the number of people. The measured values and numerical analysis results are almost identical from 09:00 to 13:00. However, from 13:00 to 15:30, there are discrepancies between the measured and the numerical analysis results. During this time, the indoor PM_2.5_ concentration increased as the children’s activity increased due to the children’s playtime. Determining indoor activity levels solely based on the set parameters presented a challenge. Future research will be necessary to predict changes in indoor PM_2.5_ concentrations with consideration for children’s activity levels.

Figure 7b presents the comparison data for Classroom 2 on 3 January 2023. Except when the children went home, the windows were closed, and the flow rates of mechanical ventilation and the air purifier were constant. Figure 7b shows that the measured values and numerical analysis results agreed. However, there is a slight discrepancy between the measured values and the numerical analysis results during the 09:30 to 09:40 and 12:30 to 13:00 timeframes. This was due to the PM_2.5_ generated by rearranging furniture for indoor classes. Unlike Figure 7a, the measured indoor PM_2.5_ concentration remains low between 13:00 and 15:00, consistent with the numerical analysis results shown in Figure 7b. This is because the activity level was low due to the children taking a nap. After 15:00, the indoor PM_2.5_ concentration increases sharply due to the inflow of PM_2.5_ from opening a window.

Figure 7c presents the comparison data for Classroom 2 on 3 January 2023. Only the flow rate of mechanical ventilation was constant, while the flow rates of the natural ventilation and air purifiers were varied. It can be seen that from 09:00 to 12:30, the measured values and numerical analysis results were generally consistent. However, from 12:30 to 15:00, there are discrepancies between the measured values and the numerical analysis results. Similar to Figure 7a, the increase in activity due to children’s indoor activity results in higher PM_2.5_ concentrations than the numerical analysis results. After 15:00, the indoor PM_2.5_ concentration increases sharply due to the inflow of PM_2.5_ from opening the windows.

Figure 7d shows the comparison data for Classroom 2 on 11 January 2023. From 09:00 to 12:00, the indoor PM_2.5_ concentration gradually increased as the outdoor PM_2.5_ concentration increased, and in the afternoon, the indoor PM_2.5_ concentration decreased as the outdoor PM_2.5_ concentration decreased. From 15:00, the window was opened, and the indoor PM_2.5_ concentration increased sharply due to the inflow of PM_2.5_. This shows that the indoor PM_2.5_ concentration is strongly affected by the outdoor PM_2.5_ concentration.

Figure 7 shows that indoor PM_2.5_ concentrations are generally affected by outdoor PM_2.5_ concentrations. In Figure 7a,b, we can see that on days with low outdoor PM_2.5_ concentrations, indoor PM_2.5_ concentrations stay below five μg/m^3^ on average. However, on days when the outdoor PM_2.5_ concentration is high, as shown in Figure 7c,d, the indoor PM_2.5_ concentration stays at an average of about 15 μg/m^3^ due to the inflow of PM_2.5_ despite the operation of the air purifier. Therefore, it is essential to reduce natural ventilation and increase the dust collection efficiency of mechanical ventilation filters on days when the outdoor PM_2.5_ concentration is high to keep the indoor PM_2.5_ concentration low.

## 4. Discussion

### 4.1. Flowrate of Mechanical Ventilation

Having validated the accuracy of Equation (2) through the previous experiment, we aimed to utilize it to present the required $Q_{MV,SA}$ based on the number of children at which CO_2_ concentrations below 1000 ppm can be maintained. Equation (2) can be simplified to Equation (6) as the term ${exp}^{-\frac{Q_{MV,SA}+Q_{inf}}{V}\;t}$ becomes negligible because the volume of most classrooms in the kindergarten is small enough to make indoor CO_2_ concentration almost be saturated within 2 h. And in Equation (6), the required $Q_{MV,SA}$ has been calculated to maintain the CO_2_ concentration below 1000 ppm.
(6)$$C_{{CO}_{2}}\left(t\right)=\frac{C_{{CO}_{2},out}\left(Q_{inf}+Q_{MV,SA}\right)+G}{Q_{exf}+Q_{MV,EA}}$$

Figure 8 shows the airflow rates supplied by mechanical ventilation to maintain indoor CO_2_ concentrations below 1000 ppm for varying numbers of children, as determined by Equation (6). The $C_{{CO}_{2},out}$ was set to 423 ppm, acquired from the annual average atmospheric CO_2_ concentration in 2021, as reported by the National Oceanic and Atmospheric Administration’s Global Atmospheric Monitoring [33]. It was assumed that two women teachers were present in the classroom. Therefore, when no children were present indoors, a minimum ventilation of 40 m^3^/h was required. If classrooms in the kindergarten are filled to maximum capacity, which is two women teachers and 15 children, a ventilation rate of 345 m^3^/h is necessary for ideal ventilation. Calculating the slope of the graph further indicates that classrooms require 20.4 m^3^/h of ventilation per child.

Figure 9 shows the days during the measurement period when the CO_2_ concentrations were highest. Furthermore, the measurements on those specific days were compared with the $Q_{MV,SA}$ calculated using Equations (2) and (6) and applied to derive the values, which was denoted as “Scenario1”. As the conditions of Scenario 1, $Q_{MV,SA}$ is 345 m^3^/h that is the highest flowrate in Figure 8, $\eta_{MV}$ is 0.35 and $\eta_{AP}Q_{AP}$ is 210 m^3^/h. Table 2 shows the values applied to the numerical model over time for Scenario 1. Figure 9a shows the results of the measurements in Classroom 1. For the measured concentration in Classroom 1, the CO_2_ concentration was almost over 1000 ppm. When Scenario 1 was applied, CO_2_ concentrations were maintained below 1000 ppm during class time. Figure 9b shows the CO_2_ concentrations measured on the same day for Classroom 2. Scenario 1, as calculated in Figure 9a, also allowed us to maintain CO_2_ concentrations below 1000 ppm.

### 4.2. Decreasing the Concentration of PM_2.5_


Figure 10 shows the measured data for the indoor PM_2.5_ concentration and the simulation under the conditions of scenarios 1, 2, and 3 on days with the highest indoor PM_2.5_ concentrations. Under the conditions of scenario 2, $Q_{MV,SA}$ is 345 m^3^/h, $\eta_{MV}$ is 0.35 and $\eta_{AP}Q_{AP}$ is 480 m^3^/h that is a level 4 flowrate for the air purifier. Under the conditions of scenario 3, $Q_{MV,SA}$ is 345 m^3^/h, $\eta_{MV}$ is 0.75 which has a MERV 13 filter grade and $\eta_{AP}Q_{AP}$ of 210 m^3^/h.

Average PM_2.5_ concentrations of a day under Scenario 1 ($Q_{MV,SA}$ = 345 m^3^/h) were about 20% higher than the measured PM_2.5_ concentration ($Q_{MV,SA}$ = 157 m^3^/h). This indicated that increasing $Q_{MV,SA}$ leads to an influx of outdoor PM_2.5_. To mitigate the increase in indoor PM_2.5_ concentrations, it is necessary to either increase the CADR ($\eta_{AP}Q_{AP}$) of air purifiers or improve the filtration efficiency of the mechanical ventilation system’s filters. Average PM_2.5_ concentrations under Scenario 2 where $\eta_{AP}Q_{AP}$ of an air purifier is increased from 210 m^3^/h to 480 m^3^/h were only about 16% lower than the measured PM_2.5_ concentration ($\eta_{AP}Q_{AP}$ = 210 m^3^/h).

In Scenario 3, the filter in the ventilation system was replaced with a MERV13 filter and the flowrate of the mechanical ventilation system was increased to 345 m^3^/h. The PM_2.5_ concentrations ($\eta_{MV}=0.75$, $Q_{MV,SA}$ = 345 m^3^/h) obtained from Scenario 3 were 52% lower than the measured PM_2.5_ concentration ($\eta_{MV}=0.35$, $Q_{MV,SA}$ = 157 m^3^/h). Even on the day that had significantly high indoor PM_2.5_ concentrations (average PM_2.5_ concentration = 26.5 μg/m^3^;), the average daily PM_2.5_ concentration can be reduced to below 15 μg/m^3^; under Scenario 3. The indoor PM_2.5_ concentration was effectively reduced by enhancing the collection efficiency of the mechanical ventilation filter when the mechanical ventilation flow rate was high enough.

## 5. Conclusions

Our study was conducted in a kindergarten setting, recognizing the heightened vulnerability of children to indoor air quality issues compared to adults. Utilizing a numerical model, we have proposed an effective method for indoor air quality management employing air purifiers and ventilation devices. Furthermore, our study introduces strategies for reducing indoor PM_2.5_ concentrations and lowering indoor CO_2_ concentration levels.

The mechanical ventilation rate necessary to maintain a CO_2_ concentration below 1000 ppm was calculated. It can be deduced that approximately 20.4 m^3^/h of ventilation per child is required for kindergarten classrooms. These results can be useful in designing a mechanical ventilation system for rooms in a kindergarten.

If a high collection efficiency filter is equipped in mechanical ventilation system, then an increased flowrate of mechanical ventilation can lead to results that show a greater reduction in indoor PM_2.5_. Consequently, a mechanical ventilation filter with a high collection efficiency for PM_2.5_ is advisable, especially in multi-use facilities such as kindergartens, where a significant amount of CO_2_ is generated via respiration.

## Figures and Tables

**Figure 1 toxics-11-00931-f001:**
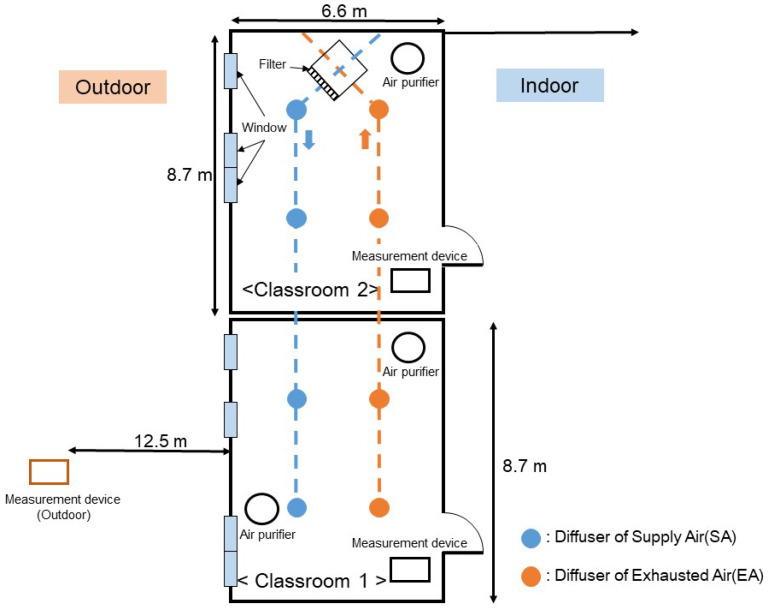
Schematic of the experimental setup: a floor plan view of the arrangement.

**Figure 2 toxics-11-00931-f002:**
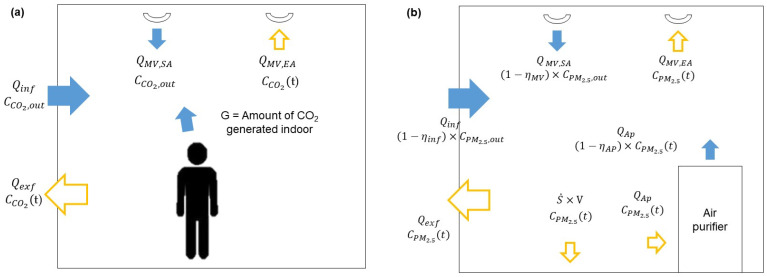
Diagrams of indoor (**a**) CO_2_ model and (**b**) PM_2.5_ model.

**Figure 3 toxics-11-00931-f003:**
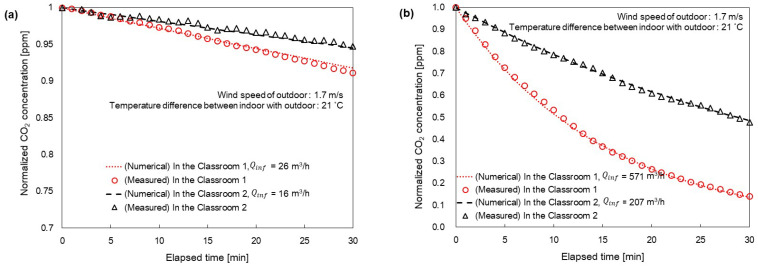
Measuring natural ventilation flow rate indoors when all windows are (**a**) closed and (**b**) opened. [Classroom 1, Classroom 2].

**Figure 4 toxics-11-00931-f004:**
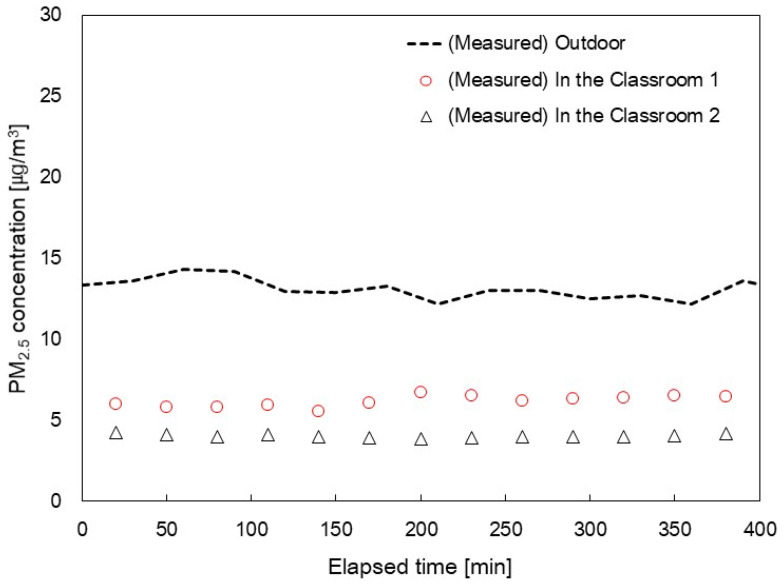
Saturated PM_2.5_ concentration indoors when an outdoor condition is stable according to time.

**Figure 5 toxics-11-00931-f005:**
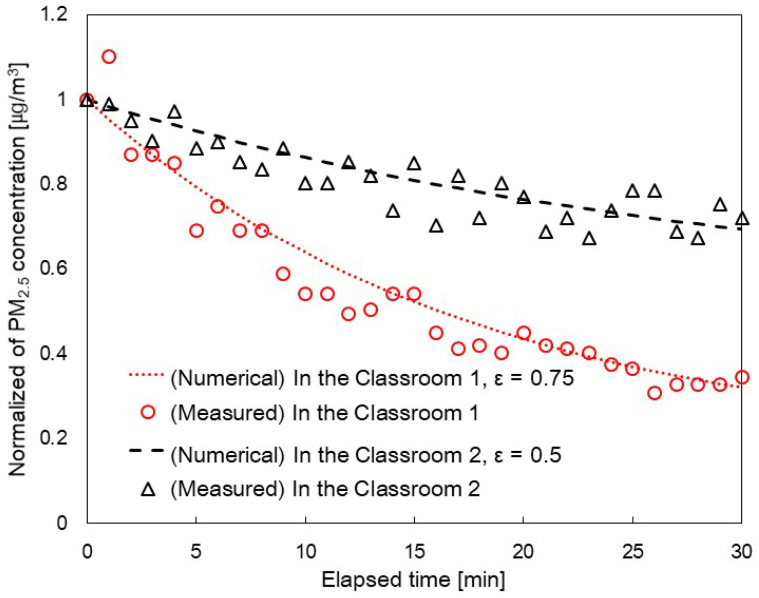
Measuring the EACR of air purifiers in Classrooms 1 and 2.

**Figure 6 toxics-11-00931-f006:**
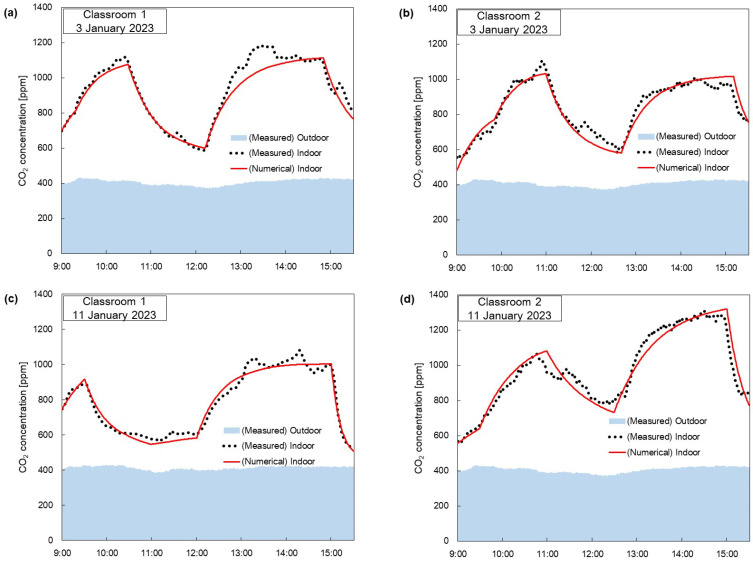
Comparison of CO_2_ concentration obtained using the numerical model in (**a**) Classroom 1 on 3 January 2023, (**b**) Classroom 2 on 3 January 2023, (**c**) Classroom 1 on 11 January 2023, (**d**) Classroom 2 on 11 January 2023 with the measured CO_2_ concentration data.

**Figure 7 toxics-11-00931-f007:**
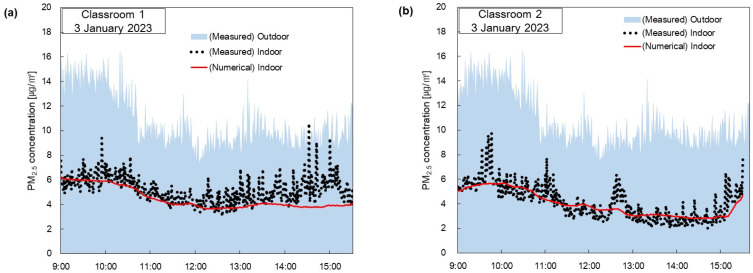
Comparison with measured data of PM_2.5_ concentration obtained by the numerical model in (**a**) Classroom 1 on 3 January 2023, (**b**) Classroom 2 on 3 January 2023, (**c**) Classroom 1 on 11 January 2023, (**d**) Classroom 2 on 11 January 2023with the measured PM_2.5_ concentration data.

**Figure 8 toxics-11-00931-f008:**
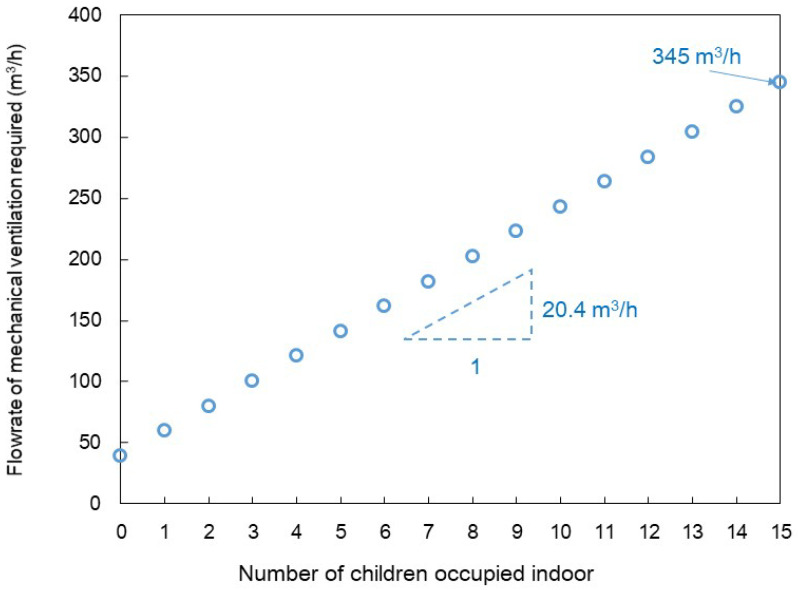
Flowrate of mechanical ventilation required, according to the number of children occupied indoor.

**Figure 9 toxics-11-00931-f009:**
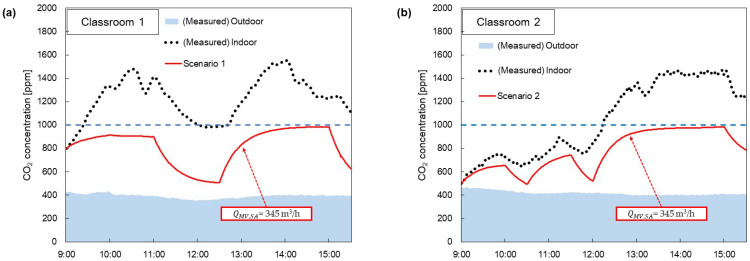
Graph showing that indoor CO_2_ concentration was maintained below 1000 ppm in ‘Scenario 1’ and that the ideal amount of ventilation was applied on days when indoor CO_2_ concentration was highest from January to March 2023 in (**a**) Classroom 1, (**b**) Classroom 2.

**Figure 10 toxics-11-00931-f010:**
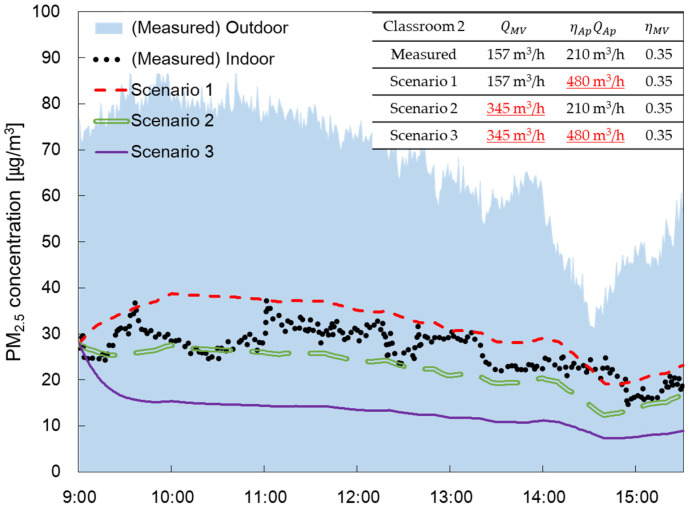
Graph comparing PM_2.5_ concentration over time obtained from simulation under the scenario conditions with measured data to show the effect of decreasing PM_2.5_ concentration.

**Table 1 toxics-11-00931-t001:** Indoor condition of CO_2_ and PM_2.5_ concentrations according to time for comparison in (**a**) Classroom 1 on 3 January 2023, (**b**) Classroom 2 on 3 January 2023, (**c**) Classroom 1 on 11 January 2023, (**d**) Classroom 2 on 11 January 2023.

Date	Classroom	Time	Number of People (Men/Women/Children)	$Q_{inf}$ (m^3^/h)	$Q_{MV,SA}$ (m^3^/h)	$\eta_{Ap}Q_{Ap}$ (m^3^/h)
(**a**)						
3 January 2023	1	9:00~9:30	1/2/8	26	230	240
		9:30~10:30	1/2/11			
		10:30~12:20	1/2/0			
		12:20~14:50	1/2/11			
		14:50~15:30	1/2/0			
(**b**)						
3 January 2023	2	9:00~9:50	0/2/7	16	254	324
		9:50~11:00	0/2/11			
		11:00~12:40	1/2/0			
		12:40~15:10	0/2/11			
		15:10~15:30	0/1/4	143		
(**c**)						
11 January 2023	1	9:00~9:40	1/2/6	26	142	120
		9:40~11:00	1/1/0	143		
		11:00~12:00		26		
		12:00~13:00	1/2/10	143		
		13:00~15:00				420
		15:00~15:30	0/2/0	571		
(**d**)						
11 January 2023	2	9:00~9:30	1/1/2	16	154	210
		9:30~11:00	1/2/7			
		11:00~12:30	1/2/0			
		12:30~15:00	2/2/8			
		15:00~15:30	1/2/0	207		

**Table 2 toxics-11-00931-t002:** Indoor condition of the Numerical Model for CO_2_ in a day when the CO_2_ concentration was highest from January to March 2023 in (**a**) Classroom 1, (**b**) Classroom 2.

(a)			
Classroom 1	Number of People (Men/Women/Children)	$Q_{inf}$ (m^3^/h)	$Q_{MV,SA}$ (m^3^/h)
9:00~11:00	1/2/11	26	142 (Measured) Indoor 345 Scenario 1
11:00~12:30	1/2/0		
12:30~15:00	0/2/15		
15:00~16:00	0/2/0		
16:00~17:30	0/2/11		
17:30~18:00	1/2/0		
(**b**)			
**Classroom 2**	**Number of People** (**Men/Women/Children**)	$Q_{inf}$ (**m^3^/h**)	$Q_{MV,SA}$ (**m^3^/h**)
9:00~10:00	0/2/4	16	154 (Measured) Indoor 345 Scenario 1
10:00~10:30	0/0/0		
10:30~11:30	0/2/8		
11:30~12:00	0/2/0		
12:00~15:00	0/2/15		
15:00~15:30	0/2/6		
15:30~16:30	0/2/13		
16:30~18:00	0/2/2		

## Data Availability

Data are contained within the article.

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
