# Peer review of "Strategies for Effective Management of Indoor Air Quality in a Kindergarten: CO2 and Fine Particulate Matter Concentrations"

_toxics, 2023, doi:10.3390/toxics11110931_

Round 1
Reviewer 1 Report
Comments and Suggestions for Authors
Dear authors,
See the attached file.

Minor editing of English language required.
Author Response
20 Oct 2023
I wish to submit the revised manuscript titled “Strategies of effective management for indoor air quality in a kindergarten: CO2 and PM2.5 concentrations”
The manuscript ID is toxics-2640488.
We thank the reviewer for your thoughtful suggestions and insights. The manuscript has benefited from these insightful suggestions. I look forward to working with the reviewer to move this manuscript closer to publication in the Toxics.
The manuscript has been rechecked and the necessary changes have been made in accordance with the reviewers’ suggestions.
Thank you for your consideration. I look forward to hearing from you.
Sincerely,
Bangwoo Han, Ph.D.
Principal Researcher
Department of Sustainable Environment Research
Korea Institute of Machinery & Materials (KIMM)
Tel: 82-42-868-7068
Fax: 82-42-868-7284
E-mail: bhan@kimm.re.kr

Reviewer 2 Report
Comments and Suggestions for Authors
Dear authors,
Your paper is exciting and the topic presented follows current trends. The article is well-written in a transparent way.
The research and its design are clear, and no significant errors were detected.
However, some additional corrections must be made for the article to be published as professional scientific work.
1. The title is straightforward and gives a sense of the topic even to an unfamiliar reader. The abstract gives a good understanding of the article and allows even a person not familiar with the field to follow. However, from reading the title and the abstract the novelty of the research is not evident. Please improve significantly.
2. Please unify the affiliation list. It also requires editing: small and capital letters used for the same names, etc.
3. The introduction is basic. The state-of-art overview of the field is very minimalistic. The authors almost straightforwardly go to the air quality in kindergartens with no real interdiction to the indoor air quality problem and different aspects of the problem. The same situation is with focusing only on the Korean region, not the problem globally. There is no mention of problems, e.g., CO2 concentration due to the chosen heating strategies and issues of selecting the right procedure in some regions due to economic situations (e.g. DOI: 10.3390/buildings13092125). No mention of different ways to monitor indoor air quality, especially novel concepts like air quality sensor data collection and analytics with IoT for ventilation systems. In the last paragraph, the novelty of the study must be explained, but it is not. This chapter must be improved.
4. Chapter “Materials and methods” – is mainly materials but no methods explained. In the case of “materials” , the information on sensors used is minimalistic without accurate parameters.
5. In the case of the figures, which are mostly well prepared, please unify the legends, captions, and texts so they look more professional. The same concern is for the tables.
6. The results presentation is well executed.
7. The conclusions are again without pointing out the novelty.
Conclusions:
The article's topic is interesting, and the paper has no significant problems with methodology, data presentation and results discussion. However, some elements are concerning. Especially the lack of evident novelty, poor introduction and explanation of materials and methods used. The paper can be considered for eventual publication after major corrections in this scope.
Author Response

(The authors gave the same response as above.)

Round 2
Reviewer 1 Report
Comments and Suggestions for Authors
I am okay with the response.
Comments on the Quality of English LanguageMinor Editing is required
Author Response
I hope this finds you well. I would like to express my sincere gratitude to you for the time and effort invested in evaluating my manuscript titled "Managing indoor air quality in a kindergarten: CO2 and PM2.5 concentrations" I greatly appreciate the valuable feedback provided, which has undoubtedly helped enhance the quality of the manuscript.
I have carefully considered all the comments and suggestions made by you and have made the necessary revisions to address your concerns. I submit a detailed response by word file.
Thank you once again for your time and consideration. I look forward to hearing from you soon regarding the status of my manuscript.

Reviewer 2 Report
Comments and Suggestions for Authors
Dear authors,
The paper was corrected. Some specific comments regarding previous elements that were pointed out can be found below.
1. The title is straightforward and gives a sense of the topic even to an unfamiliar reader. The abstract gives a good understanding of the article and allows even a person not familiar with the field to follow. However, from reading the title and the abstract the novelty of the research is not evident. Please improve significantly.
Iteration 2: This was corrected.
2. Please unify the affiliation list. It also requires editing: small and capital letters used for the same names, etc.
Iteration 2: This was corrected
3. The introduction is basic. The state-of-art overview of the field is very minimalistic. The authors almost straightforwardly go to the air quality in kindergartens with no real interdiction to the indoor air quality problem and different aspects of the problem. The same situation is with focusing only on the Korean region, not the problem globally. There is no mention of problems, e.g., CO2 concentration due to the chosen heating strategies and issues of selecting the right procedure in some regions due to economic situations (e.g. DOI: 10.3390/buildings13092125). No mention of different ways to monitor indoor air quality, especially novel concepts like air quality sensor data collection and analytics with IoT for ventilation systems. In the last paragraph, the novelty of the study must be explained, but it is not. This chapter must be improved.
Iteration 2: The introduction was partially corrected. The authors still do not present the background of the field as the main introduction and go almost straight to a specific case study of air quality in kindergartens. Please look at the initial comment regarding the introduction.
4. Chapter “Materials and methods” – is mainly materials but no methods explained. In the case of “materials” , the information on sensors used is minimalistic without accurate parameters.
Iteration 2: The chapter was partially corrected. Still not information on sensor parameters like e.g. accuracy, sensitivity etc.
5. In the case of the figures, which are mostly well prepared, please unify the legends, captions, and texts so they look more professional. The same concern is for the tables.
Iteration 2: This was corrected
6. The results presentation is well executed.
Iteration 2: No further comments.
7. The conclusions are again without pointing out the novelty.
Iteration 2: This was corrected
Conclusions:
Some elements are still of some concern. The novelty statement, poor introduction and explanation of materials and methods used are still problematic.
Author Response

(The authors gave the same response as above.)
